# MATERIALFUSION: HIGH-QUALITY, ZERO-SHOT, AND CONTROLLABLE MATERIAL TRANSFER WITH DIFFUSION MODELS

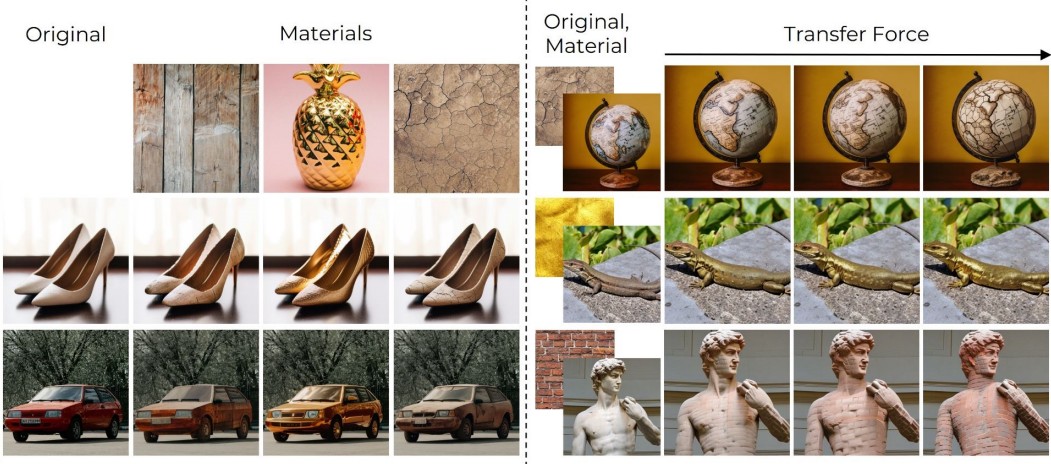

Figure 1: Examples of material transfer results using our proposed **MaterialFusion** framework. (Left) Original images and material exemplars. (Right) Progressive transfer of material properties, with increasing "transfer force" for controlled adjustments.

## ABSTRACT

Manipulating the material appearance of objects in images is critical for applications like augmented reality, virtual prototyping, and digital content creation. We present **MaterialFusion**, a novel framework for high-quality material transfer that allows users to adjust the degree of material application, achieving an optimal balance between new material properties and the object's original features. MaterialFusion seamlessly integrates the modified object into the scene by maintaining background consistency and mitigating boundary artifacts. To thoroughly evaluate our approach, we have compiled a dataset of real-world material transfer examples and conducted complex comparative analyses. Through comprehensive quantitative evaluations and user studies, we demonstrate that MaterialFusion significantly outperforms existing methods in terms of quality, user control, and background preservation. Code is available at `https://github.com/`

## 1 INTRODUCTION

Manipulating the material appearance of objects in images is a critical task in computer vision and graphics, with wide-ranging applications in augmented reality, virtual prototyping, product visualization, and digital content creation. Material transfer is the process of applying the material properties from a source exemplar to the target object in an image—enables users to visualize objects under different material conditions without the need for complex 3D modeling or rendering. This capability accelerates design workflows and enhances the realism of synthesized images, making it an important area of research.

Despite its significance, achieving high-quality material transfer remains challenging due to difficulties in preserving geometric characteristics, controlling the degree of material application, and effectively handling object boundaries. Existing methods Yeh et al. (2024); Richardson et al. (2023)

often distort the target object's shape or surface details when applying new material properties, compromising its geometric fidelity. Moreover, many approaches Sharma et al. (2023); Cheng et al. (2024); Titov et al. (2024) lack flexibility in adjusting the extent of material transfer, leading to excessive application that overwhelms the object's original structure and details, resulting in unnatural appearances. Additionally, improper blending at object boundaries can introduce noticeable artifacts, detracting from the overall image quality and disrupting consistency with the background.

Existing methods like ZeST Cheng et al. (2024) attempt to address material transfer without relying on explicit 3D information, but they often suffer from quality issues such as poor preservation of geometric characteristics and lack of control over the degree of material transfer. Furthermore, general-purpose image editing techniques, which include IP-Adapter Ye et al. (2023), Guide and Rescale Titov et al. (2024), and DreamBooth Ruiz et al. (2023), struggle with material transfer tasks. They may not accept material exemplars as input images, or if they do, they fail to produce satisfactory results, especially in preserving material properties and handling background integration.

To overcome these limitations, we propose MaterialFusion, a novel framework that combines the IP-Adapter with the Guide-and-Rescale (GaR) method within a diffusion model to achieve high-quality material transfer with enhanced control and fidelity. Our approach uses the IP-Adapter to encode material features from a source exemplar image, capturing the specific textures and nuances of the material to be transferred. Concurrently, GaR helps preserve the geometric characteristics and essential features of the target object, maintaining its original structure and details. To address issues of unintended material application and background alterations, we introduce a dual masking strategy: first, we apply masking during the material transfer process to confine the transfer to the desired regions; second, we perform masking after each denoising step to seamlessly integrate the modified object into the background and mitigate boundary artifacts. This combined approach allows for precise control over the degree and location of material transfer, resulting in natural and realistic images that maintain consistency with the surrounding environment.

Our main contributions are as follows:

- We present **MaterialFusion**, a novel framework that significantly improves the quality of material transfer in images by addressing the shortcomings of existing methods without relying on explicit 3D information.

- We introduce an adjustable material transfer control mechanism, enabling users to finetune the extent of material application. This allows for a balanced integration of new material properties with the object's original appearance, maintaining natural and realistic results.

- We have compiled an extensive dataset of real-world material transfer examples and conducted detailed comparative analyses. Through comprehensive quantitative evaluations and user studies, we demonstrate that our method outperforms existing approaches in both quality and flexibility.

## 2 RELATED WORK

**Material transferring.** Research on material transfer has progressed considerably, with early work by Khan et al. (2006) introducing methods to render objects transparent and translucent using luminance and depth maps. More recent approaches use Generative Adversarial Networks (GANs) Goodfellow et al. (2020) for high-quality material edits that adjust perceptual attributes like glossiness and metallicity, while maintaining the object's geometric structure Delanoy et al. (2022); Subias & Lagunas (2023). These GAN-based methods facilitate the modification of material appearance from a single input image, allowing for flexible and visually coherent edits.

Diffusion models have also emerged as effective tools for material modification. Sharma et al. (2024) introduced a technique using Stable Diffusion v1.5 to control material properties, including roughness, metallicity, and transparency, directly in real images. More recently, Cheng et al. (2025) proposed a model that also enables material property control; however, unlike Alchemist, their approach predicts only directions in the CLIP space. Several diffusion-based methods have been developed for 3D texturing as well, such as Text2Tex Chen et al. (2023) and TEXTure Richardson et al. (2023), which generate textures from object geometries and text prompts, and TextureDreamer Yeh et al. (2024), which transfers relightable textures to 3D shapes from a few input images. The work most similar to ours is ZeST Cheng et al. (2024), a zero-shot material transfer method that

applies exemplar materials from reference images to target objects, showcasing effective single-shot material editing without additional training.

**Diffusion Models for Image Editing.** Diffusion models have become essential in image editing, enabling detailed, high-quality transformations Ho et al. (2020); Dhariwal & Nichol (2021). Methods such as Null-text Inversion Mokady et al. (2023) and Prompt-to-Prompt Hertz et al. (2022) allow edits on real images by adjusting text prompts or modifying cross-attention layers, preserving key visual content while providing control over specific areas. InstructPix2Pix Brooks et al. (2023) extends this with instruction-driven edits, while ControlNet Zhang et al. (2023) leverages additional conditioning inputs like edge maps and segmentation masks for precise structure manipulation. However, these techniques often lack the fine-grained control needed for material-specific edits.

Self-guidance Epstein et al. (2023) and IP-Adapter Ye et al. (2023) enable image-based conditioning and layout preservation, with the Guide and Rescale (GaR) method Titov et al. (2024) further refining spatial structure by preserving attention and feature maps during edits. These methods improve detail retention but can struggle with unintended background changes and fine material control. Our approach, MaterialFusion, combines IP-Adapter's detailed material encoding with GaR's geometric fidelity and includes a dual-masking strategy to limit material transfer to targeted areas. This unified approach addresses the limitations of existing methods, ensuring high-quality, controlled material transfer while maintaining background consistency.

## 3 METHOD

In this section, we will formulate the problem statement and discuss the methods that will be employed for material transfer from one image to another. Our task involves transferring texture or material from one image $y_{im}$ into an object in the foreground of another image $x_{init}$, while preserving the background information and fine-grained details of the object. The goal is to generate an image that corresponds to the target prompt $y_{trg}$, where the object in this image is imbued with the material from $y_{im}$. The input of our model consists of an object-centric image with the target object $x_{init}$, an image of the material $y_{im}$ that can be represented as either texture or another object, and two text prompts: target prompt $y_{trg}$ and the source prompt $y_{src}$. An accompanying image (see Fig. 2) illustrates this process, highlighting the inputs and outputs of the model.

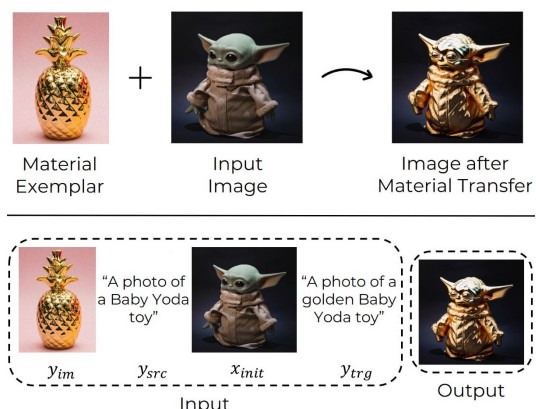

Figure 2: Overview of the material transfer process in **MaterialFusion**. Starting with a material exemplar $y_{im}$, an input image $x_{init}$, and prompts, our framework produces a target image where the object adopts the desired material properties from $y_{im}$.

To address this problem, we will employ three primary methods: the Stable Diffusion v1.5 model Rombach et al. (2022), the IP-Adapter Ye et al. (2023) for material encoding and embedding, and the Guide-and-Rescale approach Titov et al. (2024), which we will refer to as GaR. The GaR method enables guidance during the image generation process to preserve the original layout, structure, and key details of objects. This ability to maintain the image's integrity will be crucial to our material transfer tasks. Additionally, masking techniques will be utilized to enhance background preservation and facilitate effective material transfer to the designated areas of the image.

### 3.1 PRELIMINARIES

**Diffusion Model.** For our material transfer problem, we utilize the Stable Diffusion v1.5 model Rombach et al. (2022), which is a latent diffusion model (LDM). An essential aspect of the Stable Diffusion model is its use of classifier-free guidance (CFG) Ho & Salimans (2022), which allows the model to generate images conditioned on specific inputs. In contrast to classifier guidance Dhariwal & Nichol (2021), which requires a separately trained classifier to direct the sampling process towards particular targets, classifier-free guidance blends the outputs of the conditioned and unconditioned

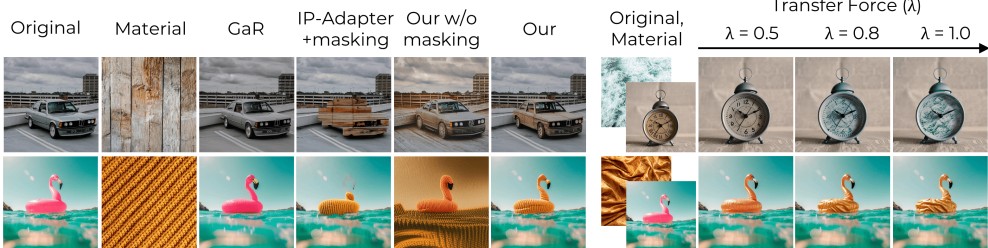

Figure 3: (Left) Comparison of material transfer results across different methods. From left to right: the original image, target material, results using Guide-and-Rescale (GaR), IP-Adapter with masking, our method without masking, and our full **MaterialFusion** approach. Our method achieves realistic material transfer while preserving object structure and background consistency. (Right) Gradual transfer of material characteristics with increasing "transfer force" ($\lambda$).

models, controlled by a guidance scale $w$. The noise prediction during the sampling stage when employing the classifier-free guidance mechanism can be mathematically expressed as:

$$\hat{\epsilon}_\theta(z_t, c, t) = w\epsilon_\theta(z_t, c, t) + (1 - w)\epsilon_\theta(z_t, t) \tag{1}$$

where $\epsilon_\theta(z_t, c, t)$ is the conditioned prediction, $\epsilon_\theta(z_t, t)$ is the unconditioned prediction and $w$ is guidance scale. This mechanism allows the model to generate high-quality outputs that are both creative and contextually aligned with the given conditions.

**Guide-and-Rescale.** In our approach to material transfer, we utilize a modified diffusion sampling process that employs a self-guidance mechanism Epstein et al. (2023), as proposed by the authors of Titov et al. (2024). The self-guidance mechanism involves leveraging an energy function $g$ to guide the sampling process, provided that a gradient with respect to $z_t$ exists.

Self-attention mechanisms, as highlighted by Tumanyan et al. (2023), effectively capture important information regarding the relative positioning of objects within an image. While the diffusion UNet layers can extract essential features from images during the forward process. Building on these insights, the authors of the GaR article developed an approach that incorporates a modified diffusion sampling process through a guidance mechanism. This enables targeted editing of specific regions within the image while preserving vital visual features—such as facial expressions—and maintaining the overall layout of the image.

First, in GaR, a DDIM inversion Song et al. (2020) trajectory is obtained $\{z_t^*\}_{t=0}^T$ for $x_{init}$, conditioning on $y_{src}$. Consequently, the single noise sampling step in GaR is defined as follows:

$$\hat{\epsilon}_\theta(z_t, c, t) = w\epsilon_\theta(z_t, c, t) + (1 - w)\epsilon_\theta(z_t, t) + v \cdot \nabla_{z_t} g(z_t, z_t^*, t, y_{src}, I^*, \overline{I}) \tag{2}$$

where $\overline{I}$ and $I^*$ are inner representations computed during the forward pass of $\epsilon_\theta(z_t, t, y_{src})$ and $\epsilon_\theta(z_t^*, t, y_{src})$ respectively, $v$ is the self-guidance scale.

**IP-Adapter.** In our study, we utilize the IP-Adapter, a lightweight and effective mechanism designed to enhance image prompt capabilities in pretrained text-to-image diffusion models. The IP-Adapter employs a decoupled cross-attention mechanism that facilitates the independent processing of text and image features. This architectural design enables the effective integration of multimodal inputs, combining both text and image prompts.

The IP-Adapter comprises an image encoder that extracts relevant features from the image prompt and adapted modules that utilize decoupled cross-attention to embed these image features into the diffusion model. Additionally, the IP-Adapter can be trained only once and then directly integrated with custom models derived from the same base model, along with existing structural controllable tools. This characteristic significantly expands its applicability and is crucial for our work, as we combine the IP-Adapter with the Guide-and-Rescale method, enhancing our capability to achieve effective material transfer.

When utilizing the IP-Adapter, the noise prediction is adapted to incorporate image conditioning, resulting in the following expression:

$$\hat{\epsilon}_\theta(z_t, c_t, c_i, t) = w\epsilon_\theta(z_t, c_t, c_i, t) + (1 - w)\epsilon_\theta(z_t, t) \tag{3}$$

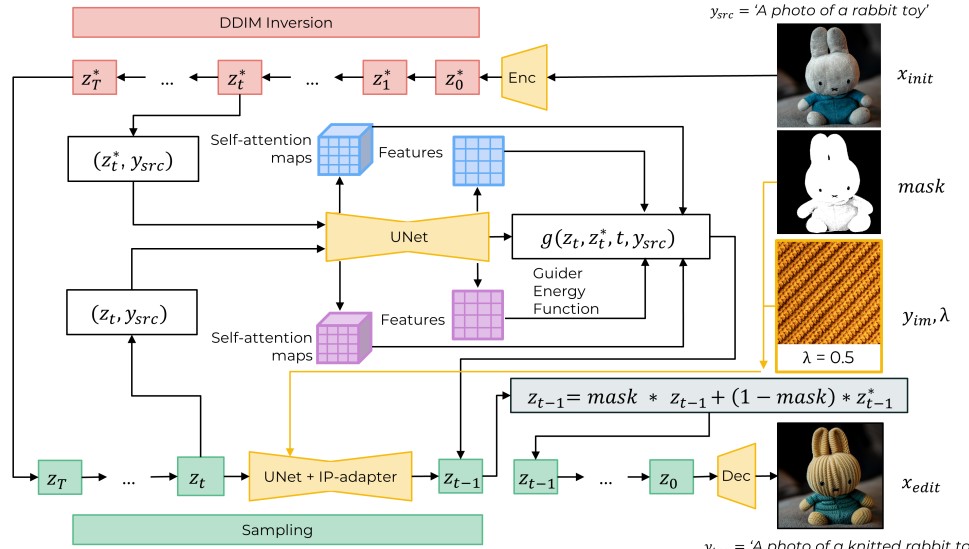

Figure 4: The overall pipeline of **MaterialFusion** for material transfer. Starting with DDIM inversion of the target image $x_{init}$ and material exemplar $y_{im}$, the framework combines the IP-Adapter with UNet and employs a guider energy function for precise material transfer. A dual-masking strategy ensures material application only on target regions while preserving background consistency, ultimately generating the edited output $x_{edit}$. The parameter $\lambda$, known as the Material Transfer Force, controls the intensity of the material application, enabling adjustment of the transfer effect according to user preference.

where $\epsilon_\theta(z_t, c_t, c_i, t)$ represents the predicted noise, $c_t$ is the text conditioning, and $c_i$ signifies the image conditioning. This formulation closely resembles the standard noise prediction seen in classifier-free guidance, but it additionally incorporates conditioning from the image prompt, enabling the generation of more contextually relevant outputs.

## 3.2 OUR METHOD

In this section, we introduce our method, which integrates the GaR approach and the IP-Adapter for effective material transfer. To begin, we will evaluate the applicability of each method independently in the context of material transfer, identifying their strengths and limitations. Understanding the challenges inherent to each approach will provide a foundation for our integrated solution.

**Guide-and-Rescale for material transfer.** In GaR, the use of a self-guidance mechanism during generation improves the editing process by preserving the initial image features and layout of the image, while the editing itself is done by CFG via a text prompt. However, relying solely on GaR proves insufficient for effectively transferring material to an object (see Fig. 3, third column). While GaR successfully retains the details of the original object, it often falls short in material transfer, resulting in either a degree of transfer that is less than desired or no transfer occurring at all. Additionally, for the task of transferring material, the strategy of changing the material via a text prompt is not suitable strategy for several reasons: firstly, generating an object with a new transferred material can be tricky for SD because the model might lean toward more typical depictions of the object. For example, generating a wooden or glass pumpkin may not be successful and could result in the generation of an ordinary typical orange pumpkin. Secondly, transferring material via text prompts requires writing large and detailed text prompts, which is not very convenient. Thirdly, text prompts can be interpreted in various ways, making it difficult to control precise attributes such as texture, color, and structural details of the material that we want to transfer.

**IP-Adapter for material transfer.** Using a text prompt to generate an object with transferred material may not yield the exact, highly specific details and nuances of the material that can be achieved by generating from a picture prompt. As the authors of the IP-Adapter article stated, "an image is worth a thousand words". All these reasons prompted us to use the IP-Adapter for encoding the material and then adding it to the target object.

The IP-Adapter consists of two main components: a pretrained CLIP Radford et al. (2021) image encoder model, which in our case extracts material-specific features from a material exemplar image, and adapted modules with decoupled cross-attention, which integrate these material features into a pretrained text-to-image diffusion model (SD v1.5 in our case).

While the IP-Adapter is a promising method for material transfer, it is important to note that it cannot independently achieve successful material transfer to an object. As shown in Fig.3 (fourth column), using the IP-Adapter to add material features to a specific region of the image via masking does not yield the desired outcomes. Although the texture of the material is transferred effectively, the object details are lost, causing them to no longer resemble their original forms. This loss of object identity is a significant issue.

**Our method, MaterialFusion as a combination of GaR and IP-Adapter.** As mentioned earlier, GaR effectively preserves the details of objects but has limitations in its ability to transfer material. Conversely, the IP-Adapter excels in material transfer but does not retain the details of the objects. To harness the advantages of both approaches, we have developed a method, which leverages the strengths of both GaR and the IP-Adapter while addressing their individual limitations.

In this combined framework, the IP-Adapter is responsible for executing the material transfer, while GaR maintains the geometry of the target object, ensuring that the background and overall pose remain intact. Additionally, GaR contributes to preserving crucial visual features of the objects, enabling a cohesive integration of the transferred material while upholding the original image details. More details on the importance of GaR in the task of material transfer are provided in the Appendix A.

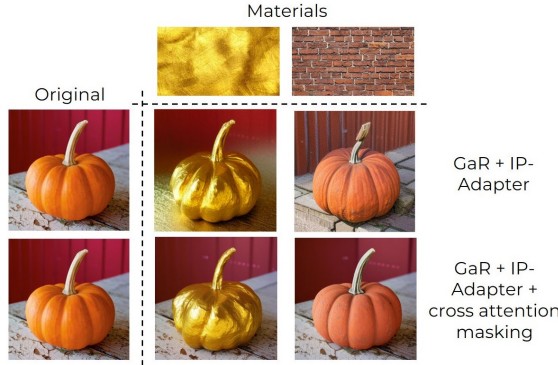

Figure 5: First Masking. After the first masking, the material is successfully transferred to the targeted area of the image. However, background preservation is not flawless, with noticeable issues occurring on the table.

The overall scheme of the proposed method, MaterialFusion, is depicted in Fig. 4. The process begins with the DDIM inversion of the source image. Subsequently, MaterialFusion conducts image editing through a denoising process, during which the UNet, in conjunction with the IP-Adapter, incorporates material features into the generated image at each denoising step. Moreover, at each step of the denoising trajectory, the noise term is adjusted by a guider that employs the latent variables $z_t$ from the current generation process, along with the time-aligned DDIM latents $z_t^*$. This adjustment helps preserve the geometry, pose, and features of the object.

A single sampling step of MaterialFusion is defined by the following formula:

$$\hat{\epsilon}_\theta(z_t, c_t, c_i, t) = w\epsilon_\theta(z_t, c_t, c_i, t) + (1-w)\epsilon_\theta(z_t, t) + v \cdot \nabla_{z_t} g(z_t, z_t^*, t, y_{src}, I^*, \overline{I}) \qquad (4)$$

where $\overline{I}$ and $I^*$ are inner representations computed during the forward pass of $\epsilon_\theta(z_t, t, y_{src})$ and $\epsilon_\theta(z_t^*, t, y_{src})$ respectively; $v$ is the self-guidance scale; $c_t$ is the text conditioning, and $c_i$ is the image conditioning. The pseudocode for the MaterialFusion method can be found in Appendix D.

**Masking for controlled Material Transfer.** Despite the initial success of our model in transferring material to target objects, we faced significant challenges, particularly regarding unintended material transfer to non-target areas and minor alterations in the background (as illustrated in the first row of Fig. 5 or in the fifth column of Fig. 3). To address these issues and enhance the precision of our approach, we implemented a masking technique for controlled Material Transfer. This technique is designed to confine the material transfer strictly to the desired regions of the object and better preserve the background.

In our method, we apply masking twice. The first masking is performed at the stage of incorporating material-specific features from a material exemplar image into a pretrained text-to-image diffusion model, which occurs through the image features cross-attention layers of IP-Adapter (see Fig. 4). This masking solves the problem of unintended material transferring to non-target areas. The results of the generation after cross-attention masking can be seen in Fig. 5, where it is evident that after this

Figure 6: To compare the qualitative results obtained by different methods: Our method, ZeST, GaR, and IP-Adapter with masking. Our method demonstrates more realistic material integration, preserving object structure and achieving higher fidelity to the target material.

masking, the material successfully transfers to the intended region, although some slight background changes can be observed.

In the second masking step, we solve the problem of background changes compared to the original image during generation. Masking is performed as follows: after each denoising step, we extract the masked object from the sampling trajectory and a masked background from the DDIM inversion latent corresponding to the current step. In other words, this can be expressed as a formula:

$$z_t = mask \cdot z_t + (1 - mask) \cdot z_t^* \tag{5}$$

where $z_t$ is the latent representation from the current generation process, $z_t^*$ is the time-aligned DDIM latent, and $mask$ is the binary mask of the object.

This formula illustrates how we combine the current latent $z_t$ and the time-aligned DDIM latent $z_t^*$ using a binary mask. The values corresponding to the object are retained from the current generation step, while the background is updated by combining with the time-consistent latent representation from the corresponding inversion step. In this way, we ensure stability and continuity of the background, avoiding abrupt transitions and artifacts. Moreover, this approach not only improves the visual quality of the final image, but also promotes a smoother integration of elements in the image, creating a lively and harmonious composition. We also determined the appropriate denoising step up to which masking should be performed. See Appendix B for details.

**Material Transfer Force.** As previously mentioned, our method employs a decoupled cross-attention mechanism from the IP-Adapter for material transfer, utilizing query features $Z$, text features $c_t$, and image features from the material exemplar $c_i$. The relevant matrices for the attention operations are defined as follows:

- For text features: $Q = ZW_q$, $K = c_tW_k$, $V = c_tW_v$
- For material image features: $Q = ZW_q$, $K' = c_iW_k'$, $V' = c_iW_v'$

Thus, the overall output of the attention mechanism is:

$$Z_{new} = Attn(Q, K, V) + \lambda \cdot Attn(Q, K', V') \tag{6}$$

Here, $\lambda$ represents the Material Transfer Force, which controls the intensity of material transfer in the output. Adjusting $\lambda$ allows for modulating the influence of material characteristics while preserving details, resulting in a coherent and visually appealing integration. As illustrated in Fig. 3, variations in $\lambda$ demonstrate the resulting effects on material transfer and detail preservation. For more details on the Material Transfer Force, please refer to the Appendix K.

## 4 EXPERIMENTS

To compare MaterialFusion with other methods, we created a dataset of free stock images comprising 15 material images and 25 object-oriented photographs. Detailed dataset description can be found in Appendix C.

We compared our method against the following approaches: Guide-and-Rescale, IP-Adapter with masking, and ZeST. We utilized the authors' original code with the default parameters specified in each method's description. Configurations of our method and the baselines are in Appendix H.

Our quantitative analysis involved an assessment of the following aspects: firstly, we focused on the preservation of the background of the images, the geometry of the objects, and the details they contain. To evaluate this, we calculated the LPIPS Zhang et al. (2018) between the original object images and those obtained through various material transfer methods.

Secondly, we aim to assess how effectively material can be transferred. To accomplish this, we developed the following scheme: we extracted crops of two sizes, 64x64 and 128x128 pixels, from the resulting images using the object mask, ensuring that only the transferred texture crops were included—without any background. Similarly, we generated crops from the example material images. Subsequently, we computed pairwise CLIP similarity scores between these crops to determine the degree of similarity between the textures and then we calculated the average of these scores. For a more comprehensive description of the metrics, see Appendix I.

It is also important to mention, that during image generation, accurately describing the material in $y_{trg}$ can be challenging. In such cases, we took $y_{trg} = y_{src}$. The reasoning behind this decision is discussed in Appendix J.

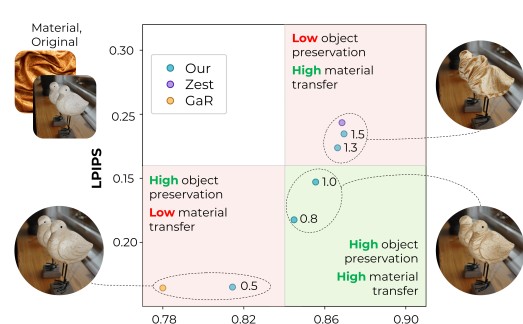

Figure 7: Quantitative analysis of material transfer and object preservation. The lower right region represents optimal results with high CLIP scores (effective transfer) and low LPIPS values (good detail preservation). The numbers above the dots in the graph represent the material transfer force. **MaterialFusion** achieves the best balance, with results in the optimal zone, while GaR and ZeST show trade-offs between transfer efficiency and detail preservation.

### 4.1 QUALITATIVE COMPARISON

Fig. 6 presents examples of material transfers utilizing various methods: ZeST, GaR, IP-Adapter with masking, and our proposed approach. The images clearly demonstrate that GaR results in minimal material transfer. While the IP-Adapter successfully captures the texture of the material, it completely fails to preserve details. ZeST consistently performs well in terms of material transfer but struggles to maintain object details. In contrast, our method exhibits robust performance in both material transfer and detail preservation. More visual comparisons are in the Appendix E.

### 4.2 QUANTITATIVE COMPARISON

Fig. 7 shows the results of our quantitative analysis. The optimal region, located in the lower right corner, indicates that a high CLIP similarity score corresponds to effective material transfer, while a low LPIPS value reflects good preservation of the object's details and image's background.

Upon examining the generated images, we find that a CLIP similarity score below $0.82$ indicates the ineffective material transfer, while a score above $0.84$ suggests successful transfer. Additionally, we noted that when the LPIPS value exceeds $0.21$, the material starts to lose its details significantly. Consequently, we have outlined the approximate region of effective material transfer combined with satisfactory preservation of the object in green on the graph. As illustrated, only two points fall within

Table 1: User study results comparing our method with ZeST. Our method was preferred overall and rated highly for detail preservation, while ZeST scored better for material fidelity. This balance between material transfer and object fidelity makes our method more effective in delivering coherent and lifelike results.

| Questions | Results |
|---|---|
| Overall Preference (Q1) | 68% |
| Material Fidelity (Q2) | 26% |
| Detail Preservation (Q3) | 70% |

this favorable zone: MaterialFusion with material transfer strengths of $0.5$ and $0.8$. The results of GaR fall into the region indicating good detail preservation but with low material transfer effectiveness. In contrast, the ZeST performs well in transferring material but fails to preserve the details.

This analysis underscores the trade-offs between material transfer efficacy and detail preservation across different methods. For a more comprehensive view, refer to Appendix F of the supplement, where Fig. 7 is expanded to include additional methods: our approach without masking and the IP-Adapter with masking.

### 4.3 USER STUDY

To evaluate the effectiveness of our method, we conducted a user study comparing our approach with ZeST, the current state-of-the-art method for material transfer. By presenting the results of both our method and ZeST, we asked participants three questions: the first question (Q1) assessed user preferences regarding Overall Preference, the second question (Q2) focused on Material Fidelity, and the third question (Q3) evaluated Detail Preservation of the image results produced by both methods. Details of the user study and all questions used can be found in Appendix G.

We conducted a user study with 100 participants, each completing 11 trials. In each trial, participants viewed four images: the target object, a reference material, the ZeST transfer result, and our method's result, yielding 1,125 responses per question.

The results of the user study are presented in Table 1. Each value indicates the percentage of users who preferred our method compared to ZeST. According to the respondents, our method produces more realistic images and better preserves the details of the original object compared to ZeST by a wide margin. However, the results of the user study indicate that we transferred the material less effectively than ZeST. There is a very simple and logical explanation for this. When using the simplest approach of cutting the material using a mask and pasting it onto the original image, the material transfer appears perfect; however, this method sacrifices any preservation of the original object. ZeST lacks control over the material transfer force, which can result in outputs that resemble the simplistic cut-and-paste technique (see Fig. 8). Therefore, the outcome is quite commendable, as it reflects a balance between maintaining the integrity of the

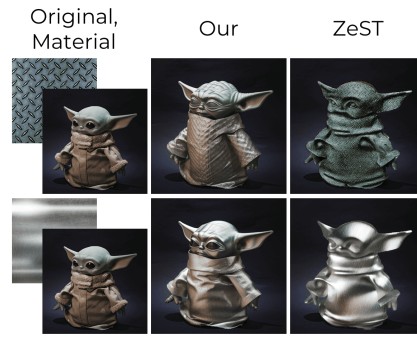

Figure 8: Examples of comparisons where the vote for Question Q2 (Material Fidelity) was given to ZeST. While ZeST achieves high material transfer, it often overpowers the original object's features, resulting in a "cut-and-paste" effect. Our approach, by contrast, balances material integration with preservation of the object's details, offering a more coherent and realistic result.

original object and achieving material transfer. Our method may not have transferred the material as effectively as ZeST, but it provided a more realistic and coherent integration of materials and original details, which is a significant achievement in its own right.

### 5 LIMITATIONS

Despite the strengths of our method, several limitations must be acknowledged. One specific limitation we have identified is that our method may encounter difficulties when transferring material to composite objects that consist of different parts made from diverse materials. Moreover, the model may struggle with material transfer when there is a significant mismatch between the material's nature and the object's nature. For instance, transferring foaming waves to a motorcycle is challenging due to their differing characteristics: foaming waves

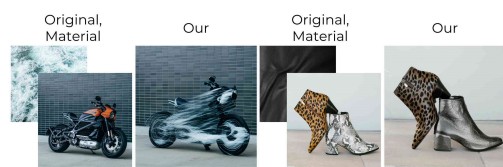

Figure 9: Limitations of Our Method. (Left) The difference in the nature of materials leads to poor transfer results. (Right) The shoe material differences resulted in limited transfer to the left shoe.

are dynamic and fluid, lacking solidity, while a motorcycle features sharp edges, flat surfaces, and well-defined shapes. This fundamental disparity leads to unrealistic transfer results (see Fig. 9).

### 6 CONCLUSION

We introduce **MaterialFusion**, a novel framework for exemplar-based material transfer that balances material fidelity with detail preservation, leveraging existing pre-trained models like IP-Adapter and Guide-and-Rescale within a unified diffusion model approach. Through quantitative evaluations and user studies, our method demonstrates superior results in realistic material integration compared to existing approaches. However, our framework has limitations, particularly in handling highly complex materials or intricate textures where fine-grained control may still fall short. Despite these challenges, MaterialFusion offers a robust foundation for future advancements in controlled, high-quality material transfer for real-world applications.

## 7 REPRODUCIBILITY STATEMENT.

Our framework does not require training. All details regarding inference, including hyperparameters and architectural descriptions, can be found in the supplementary materials (Sections B, H, and J). All experiments and the methods for calculating metrics are detailed in the text, particularly in Section 4 of the main text and Sections C–K of the supplementary material. Furthermore, the code for our method is available on GitHub. Since our experiments rely solely on our custom dataset, which is publicly available, all reported results can be fully reproduced.

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

## A  Necessity of guiders

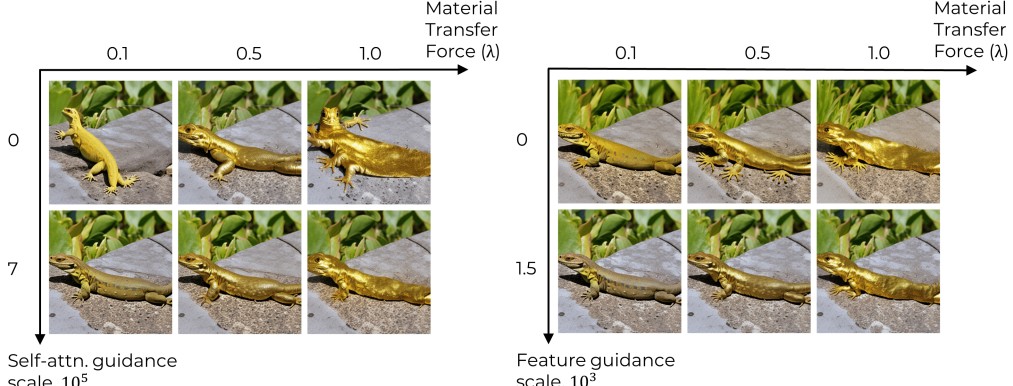

Figure 10: Illustration of the significance of using guiders: (Left) Self-attention guider; (Right) Feature guider. Transferring material without these guiders fails to maintain the object's geometry, visual features, and pose.

As previously mentioned, the Guide-and-Rescale method employs an energy function $g$ to enhance the sampling process. The authors of the GaR approach introduced two guiding mechanisms: the Self-attention Guider and the Feature Guider. When utilized together during the generation process, these guiders significantly enhance the preservation of original image details.

Self-attention mechanisms, as noted by the authors of Tumanyan et al. (2023), capture significant information regarding the relative positioning of objects within an image. So authors of GaR suggested guiding through matching of self-attention maps from the current trajectory $\overline{A}_i^{\text{self}} := \text{selfattn.}[\epsilon_\theta(z_t, t, y_{\text{src}})]$ and an ideal reconstruction trajectory $A_i^{*\text{self}} := \text{selfattn.}[\epsilon_\theta(z_t^*, t, y_{\text{src}})]$, where $i$ corresponds to the index of the UNet layer. So the self-attention guider is defined as follows:

$$g_{\text{self}}(z_t, z_t^*, t, y_{\text{src}}, \{A_i^{*\text{self}}\}, \{\overline{A}_i^{\text{self}}\}) = \sum_{i=1}^{L} \text{mean}||A_i^{*\text{self}} - \overline{A}_i^{\text{self}}||_2^2 \tag{7}$$

Moreover, during the forward process, diffusion UNet layers can extract essential features from images. In GaR authors defined features $\Phi$ as an output of the last up-block in UNet. If $\overline{\Phi} = \text{features}[\epsilon_\theta(z_t, t, y_{\text{src}})]$ and $\Phi^* = \text{features}[\epsilon_\theta(z_t^*, t, y_{\text{src}})]$ than feature guider is defined as:

$$g_{\text{feat}}(z_t, z_t^*, t, y_{\text{src}}, \Phi^*, \overline{\Phi}) = \text{mean}||\Phi^* - \overline{\Phi}||_2^2 \tag{8}$$

In our approach, we combine both the self-attention guider and the feature guider to maintain the layout, visual features, and geometry. Specifically, in our task of material transfer, the self-attention guider is primarily responsible for preserving the geometry and pose of the target object. Meanwhile, the feature guider focuses on maintaining the visual characteristics of the object. Although the feature guider also contributes to preserving the geometry, its effectiveness in this regard is somewhat less than that of the self-attention guider.

Thus, a single sampling step in MaterialFusion can also be expressed as follows:

$$\hat{\epsilon}_\theta(z_t, c_t, c_i, t) = w\epsilon_\theta(z_t, c_t, c_i, t) + (1 - w)\epsilon_\theta(z_t, t) +$$
$$+\gamma[v_{\text{self}} \cdot \nabla_{z_t} g_{\text{self}}(z_t, z_t^*, t, y_{src}, \{A_i^{*self}\}, \{\overline{A}_i^{self}\}) +$$
$$+v_{\text{feat}} \cdot \nabla_{z_t} g_{\text{feat}}(z_t, z_t^*, t, y_{src}, \Phi^*, \overline{\Phi})] \tag{9}$$

Here, $\gamma$ is a scaling factor (the method of calculating $\gamma$ is detailed in the Guide-and-Rescale article), and $v_{\text{self}}$ and $v_{\text{feat}}$ represent the self-guidance scale and feature-guidance scale, respectively.

By adjusting these scales, one can modulate the influence of the guiders on the generated output. The significance of employing both the self-attention guider and the feature guider is illustrated in Fig. 10.

## B   ANALYSIS OF MASKING IN DENOISING PROCESSES

As mentioned earlier, we also identified the appropriate denoising step up to which the second masking step—adding background from the DDIM inversion trajectory—should be executed.

As shown in Fig. 11, masking during the early iterations fails to preserve the image background, while masking across the entire denoising trajectory helps maintain the background but negatively impacts the quality of the generated output. To balance these effects, we chose an intermediate value of 40 out of 50 denoising steps for masking. This approach allows us to achieve both high generation quality and effective background preservation.

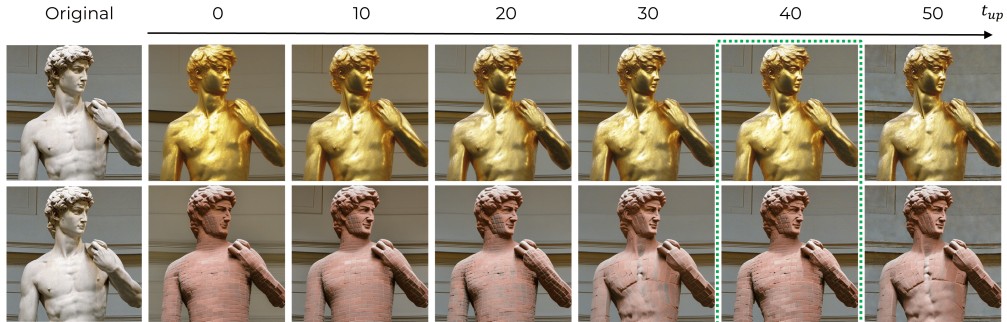

Figure 11: Identification of the optimal denoising step for executing the second masking — adding background from the DDIM inversion trajectory. By masking the first 40 out of 50 denoising steps in this manner, we effectively preserve the background while achieving high-quality generation

## C   DATASET DESCRIPTION

To compare MaterialFusion with other methods, we created our own dataset of real free stock images. Our dataset comprises 15 material images and 25 object-oriented photographs of various objects. Fig.12 showcases examples of objects and materials from the dataset.

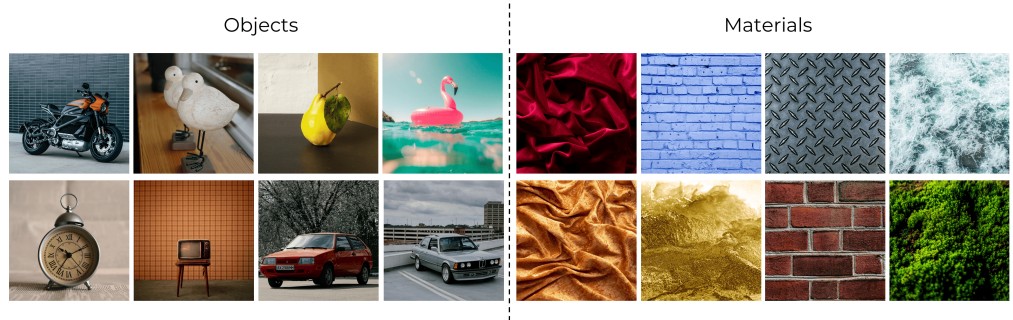

Figure 12: Examples of images from our custom dataset: (Left) Object images; (Right) Material images.

## D   PSEUDOCODE FOR THE PROPOSED METHOD

The proposed method is summarized in the Algorithm 1.

---

**Algorithm 1** MaterialFusion

---

**Input:** Real image $x_{\text{init}}$, source prompt $y_{\text{src}}$, target prompt $y_{\text{trg}}$, material exemplar image $y_{\text{im}}$; DDIM steps $T$; guidance scales $w, v_{\text{self}}, v_{\text{feat}}$; guidance threshold $\tau_g$; masking threshold $\tau_m$; noise rescaling boundaries $r_{\text{lower}}, r_{\text{upper}}$; material transfer force $\lambda$; binary object mask $mask$.

**Function:** VAE encoder $Enc.$, VAE decoder $Dec.$, DDIM Inversion Song et al. (2020), DDIM Sample Song et al. (2020), Self-attention Guider $g_{\text{self}}$ (Equation 7), Feature Guider $g_{\text{feat}}$ (Equation 8 ), noise rescaling $f_\gamma$ Titov et al. (2024).

**Output:** Edited image $x_{\text{edit}}$.

1: $z_0^* = Enc.(x_{\text{init}})$
2: **for** $t = 0, 1, \ldots, T - 1$ **do**
3:     $z_{t+1}^* = \text{DDIM Inversion}(z_t^*, y_{\text{src}})$
4: **end for**
5: $z_T = z_T^*$
6: **for** $t = T, T - 1, \ldots, 1$ **do**
7:     $\Delta_{\text{cfg}} = w(\varepsilon_\theta(z_t, t, y_{\text{trg}}, y_{\text{im}}, \lambda, mask) - \varepsilon_\theta(z_t, t, \varnothing))$
8:     $\epsilon_{\text{cfg}} = \varepsilon_\theta(z_t, t, \varnothing) + \Delta_{\text{cfg}}$
9:     $\{\{\mathcal{A}_i^{*\text{self}}\}_{i=1}^L, \Phi^*\} = \varepsilon_\theta(z_t^*, t, y_{\text{src}})$
10:    $\{\{\bar{\mathcal{A}}_i^{\text{self}}\}_{i=1}^L, \bar{\Phi}\} = \varepsilon_\theta(z_t, t, y_{\text{src}})$
11:    $\epsilon_{\text{self}} = v_{\text{self}} \cdot g_{\text{self}}(\{\mathcal{A}_i^{*\text{self}}\}_{i=1}^L, \{\bar{\mathcal{A}}_i^{\text{self}}\}_{i=1}^L)$
12:    $\epsilon_{\text{feat}} = v_{\text{feat}} \cdot g_{\text{feat}}(\Phi^*, \bar{\Phi})$
13:    $r_{\text{cur}} = \|\Delta_{\text{cfg}}\|_2^2 / \|\nabla_{z_t}(\epsilon_{\text{self}} + \epsilon_{\text{feat}})\|_2^2$
14:    $\gamma = f_\gamma(r_{\text{lower}}, r_{\text{upper}}, r_{\text{cur}})$
15:    **if** $T - t < \tau_g$ **then**
16:       $\epsilon_{\text{final}} = \epsilon_{\text{cfg}} + \gamma \cdot \nabla_{z_t}(\epsilon_{\text{self}} + \epsilon_{\text{feat}})$
17:    **else**
18:       $\epsilon_{\text{final}} = \epsilon_{\text{cfg}}$
19:    **end if**
20:    $z_{t-1} = \text{DDIM Sample}(z_t, \epsilon_{\text{final}})$
21:    **if** $T - t < \tau_m$ **then**
22:       $z_{t-1} = mask \cdot z_{t-1} + (1 - mask) \cdot z_{t-1}^*$
23:    **end if**
24: **end for**
25: $x_{\text{edit}} = Dec.(z_0)$
**Return:** $x_{\text{edit}}$

---

# E  ADDITIONAL VISUAL COMPARISON

In this section, we present an additional visual comparison of our method against ZeST, IP-Adapter with masking, and GaR, as illustrated in Fig.14. The results indicate that while GaR demonstrates a strong capability to maintain visual features, it struggles with effective material transfer. Conversely, IP-Adapter with masking is proficient at transferring material textures but often compromises the preservation of the objects' underlying features. ZeST performs well in transferring materials for simple objects, such as chairs, yet it falls short in maintaining features when dealing with more complex objects. In contrast, our method effectively transfers materials to complex objects while maintaining their visual features.

Additionally, Fig.18 provides a direct comparison between our method and the current state-of-the-art method ZeST, highlighting the improvements achieved by our approach.

# F  EXTENDED QUANTITATIVE ANALYSIS

Figure 13 provides an enhanced version of Figure 7, introducing two additional methods: the IP-Adapter with masking and our method without masking. The material transfer force, represented by the numbers above the dots in the graph for the methods—Ours, IP-Adapter with masking, and our method without masking—was varied between $0.5$ (indicating weak material transfer) and $1.5$ (indicating excessively strong material transfer). The results indicate that all three methods improve material transfer effectiveness as the material transfer force increases, as evidenced by the rise in the

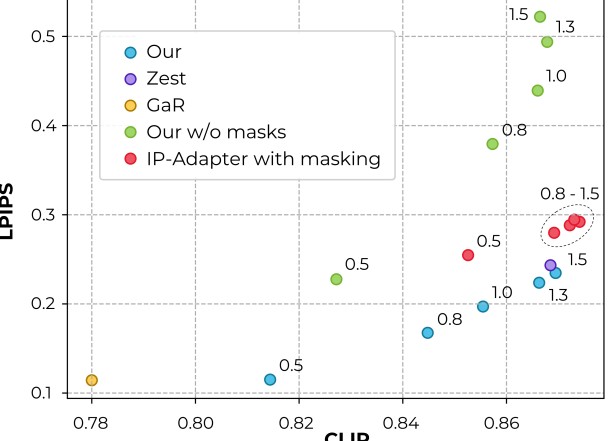

Figure 14: Qualitative comparison with baselines, including ZeST, Guide-and-Rescale, and IP-Adapter with masking, to integrate material features into specific regions of the image.

CLIP similarity score. However, this enhancement comes at the expense of detail retention, as illustrated by the increasing LPIPS scores. Notably, at a material transfer force of 1.5, the performance metrics of our method closely resemble those of ZeST.

The graph also reveals that our method without masking leads to a significant increase in LPIPS compared to our masked approach, indicating that masking is crucial for preserving background details while effectively transferring material to the intended areas of the image.

Additionally, it is evident from the graph that the IP-Adapter with masking, despite enhancing material transfer relative to our method, fails to retain object details, as indicated by the high LPIPS scores.

## G    USER STUDY

In this appendix, we present more details on the user study conducted to evaluate the effectiveness of our proposed method. In our study, each respondent was presented with a set of four images: the original object, an example image of the material to be transferred, the result generated by ZeST, and the result produced by our proposed method. Participants were asked to answer three specific questions:

**Q1:** Which image do you prefer? Assess the overall quality of the image: are details added or removed, is the image spoiled (e.g., noise, blurriness), and is it realistic?

**Q2:** Which image better transfers the features of the material? Can we say that the object is now made of this material or that it uses this material?

Figure 13: Extended quantitative analysis of material transfer and object preservation. The numbers above the dots in the graph represent the material transfer force for the following methods: our method, IP-Adapter with masking, and our method without masking.

**Q3:** Which image better preserves the original object, including its outlines, details, and depth?

## H  DETAILED CONFIGURATION OF THE METHOD AND BASELINES

All experiments comparing the methods were performed using the official repositories from the authors. The relevant code implementations and specific parameters for the method's inference, including those for our method, are listed in Table 2. All experiments were conducted on a NVIDIA V100 with 40GB RAM.

Table 2: Overview of material transfer methods. Table presents the methods employed for material transfer, along with their respective implementations and configuration settings. Source repositories are included for reference.

| Method | Used Implementation | Configuration Settings |
|---|---|---|
| ZeST | ZeST github-repo | N/A |
| IP-Adapter + masking | IP-Adapter github-repo | $\tau_m = 40$ |
| Guide-and-Rescale | GaR github-repo | $w = 7.5$, $\tau_g = 30$, $v_{\text{self}} = 300000, v_{\text{feat}} = 500$, $r_{\text{lower}} = 0.33, r_{\text{upper}} = 3$ |
| Our | - | $w = 7.5$, $\tau_g = 30, \tau_m = 40$, $v_{\text{self}} = 700000, v_{\text{feat}} = 1500$, $r_{\text{lower}} = 0.33, r_{\text{upper}} = 3$ |

## I  DESCRIPTION OF EVALUATION METRICS

In this appendix, we detail the metrics employed in our quantitative analysis, along with the calculation methods used.

**Learned Perceptual Image Patch Similarity (LPIPS).**  Learned Perceptual Image Patch Similarity (LPIPS) is utilized to assess the perceptual similarity between the original object images and those generated through various material transfer methods. This metric is crucial for our analysis, as it allows us to evaluate the preservation of the background, object geometry, and intricate details within the images. LPIPS computes the similarity between the activations of two image patches based on a pre-defined neural network.

For our analysis, we employ AlexNet for feature extraction, which as a forward metric, according to the LPIPS GitHub documentation, performs best. The process involves the following steps:

1. **Feature Extraction:** LPIPS extracts features from the original and generated images using the activations from specific layers of AlexNet, which captures crucial perceptual information about the images.

2. **Similarity Computation:** By comparing the extracted feature activations from the two images, LPIPS quantifies how similar they are in terms of perceptual content.

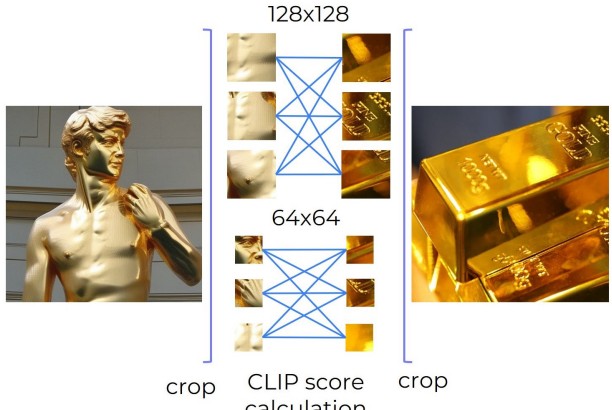

Figure 15: CLIP-based similarity scoring using 64x64 and 128x128 crops of material-transferred and sample images, excluding background. Pairwise scores quantify texture similarity.

A lower LPIPS score indicates high similarity between the original and generated images, while a higher score signifies greater divergence.

3 **Average Calculation:** Next, for each material transfer method, the average LPIPS score is calculated across all images in the dataset.

**CLIP similarity score.** To evaluate the effectiveness of material transfer, we utilize the CLIP similarity score. Calculating the similarity between two images using CLIP involves two main steps: first, we extract the features of both images, and then we compute their cosine similarity. A detailed explanation of the CLIP similarity score calculation within the context of our material transfer task can be found in the "Experiments" section of the article. This process is also illustrated in Fig.15.

## J ABLATION STUDY ON THE NECESSITY OF TARGET PROMPT $y_{trg}$

Our primary objective is to generate an image that corresponds to the target prompt $y_{trg}$, where the object depicted in this image acquires the material properties from $y_{im}$. Notably, it is feasible to utilize $y_{src}$ in place of $y_{trg}$, particularly in instances where the material characteristics are challenging to articulate. This approach allows for effective material transfer using solely the reference image $y_{im}$.

In our ablation study, depicted in Fig.16, we investigate whether the material description in the target prompt is essential for effective material transfer. We compare the results of material transfer when the target prompt $y_{trg}$ includes a material description versus scenarios where the target prompt is set as $y_{trg} = y_{src}$ (without a specific material description). Using four distinct pairs of objects and materials, we find that material transfer occurs successfully in all cases, irrespective of the presence of a material description in the target prompt. Notably, for certain pairs—such as Baby Yoda with gold, pears with masonry, and pears with leather—the inclusion of a material description allows for the transfer to commence at lower force levels compared to cases without such a description. Conversely, for other pairs, like the rabbit toy with knitted fabric, the presence of a material description does not influence the material transfer process at all, maintaining consistent transfer forces across both conditions.

## K MATERIAL TRANSFER FORCE

In this appendix, we provide a collection of examples illustrating the concept of material transfer force, as shown in Fig.17. As seen in the image, the process begins with the transfer of texture from the material exemplar, followed by the transfer of color. This sequential representation highlights the distinct phases of material transfer.

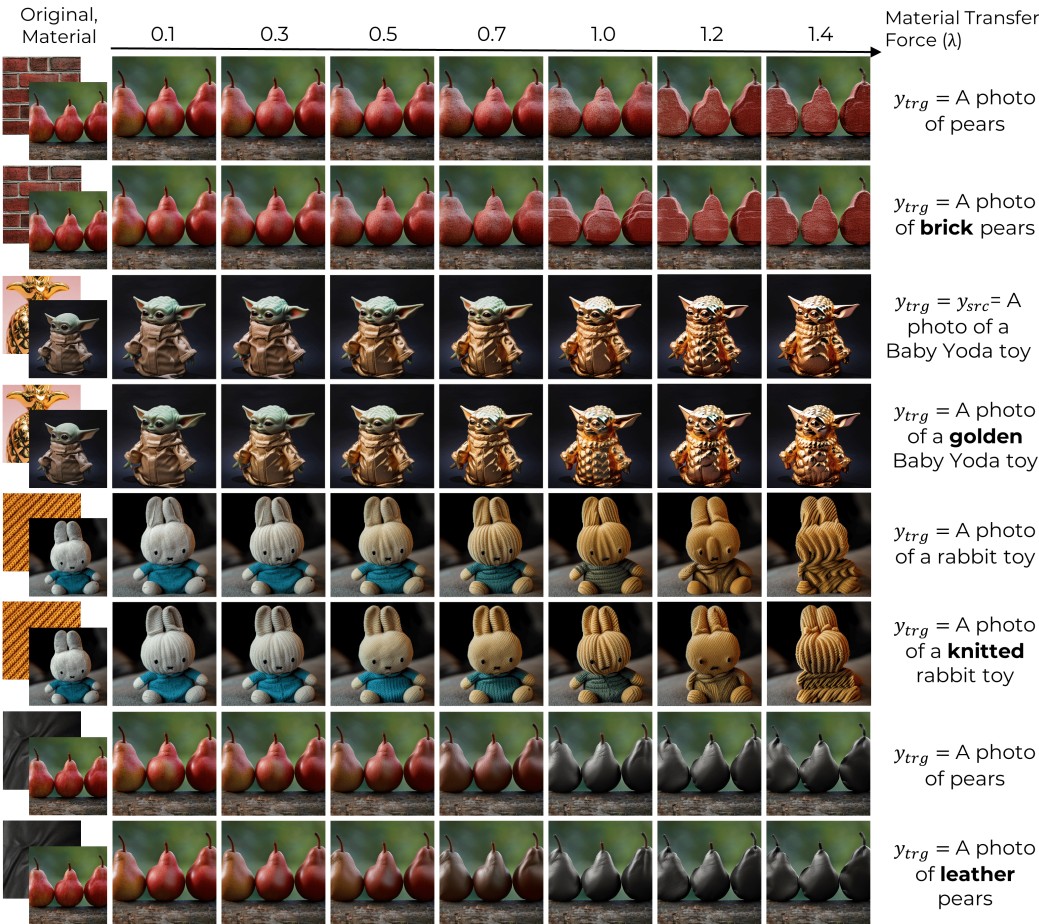

Figure 16: Ablation study on the necessity of the target prompt $y_{trg}$. We demonstrate that material transfer remains successful regardless whether the target prompt is present or not (i.e., when $y_{trg} = y_{src}$). However, we observed that the presence of the target prompt $y_{trg}$ facilitates easier material transfer, resulting in lower material transfer forces.

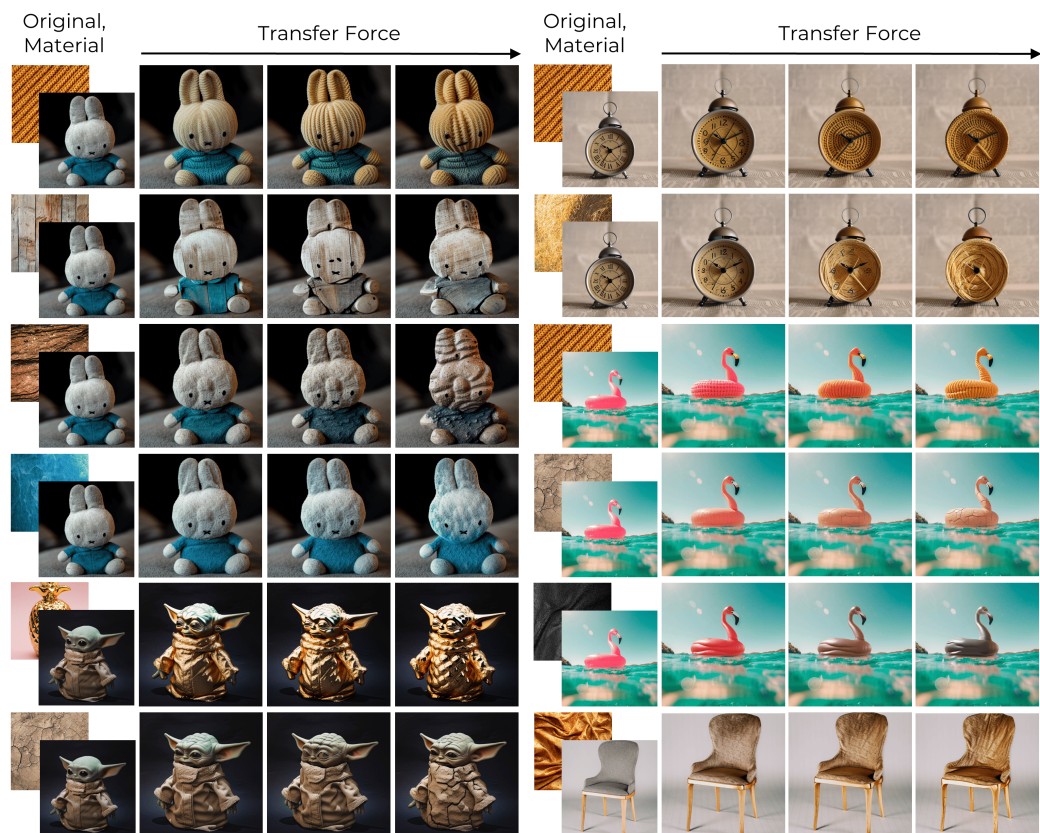

Figure 17: Examples of controlled addition of material to an object. The increase in material transfer force can result in various outcomes, including changes in physical properties as well as modifications to texture and color. By maintaining precise control over the material transfer process, these modifications can be carefully implemented, ensuring that the desired characteristics are achieved without compromising the object's original design.

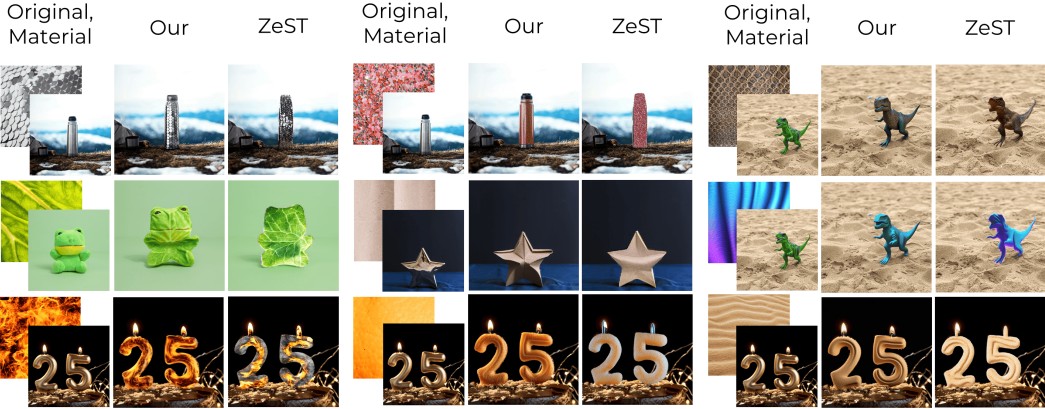

Figure 18: Direct comparison between our method and the current state-of-the-art ZeST. Our approach effectively transfers materials to complex objects while better preserving their visual features compared to ZeST.

