# OpenReview forum: "MaterialFusion: High-Quality, Zero-Shot, and Controllable Material Transfer with Diffusion Models"
_ICLR.cc/2026/Conference — ICLR 2026 Conference Withdrawn Submission_

### Official Review · Reviewer_bJ5r · 2025-10-25

**Soundness:** 2
**Presentation:** 3
**Contribution:** 2
**Rating:** 2
**Confidence:** 4

**Summary:**

MaterialFusion is a framework for material transfer that transfers material appearance from an exemplar image to a masked region in a target image. It integrates prior work—IP-Adapter for exemplar encoding and Guide-and-Rescale (GaR) for structural preservation—to achieve a balance between material fidelity and object detail retention. The paper also introduces a dual-masking strategy to better preserve background regions. Furthermore, users can control the transfer strength via a Material Transfer Force parameter, which is a weighting factor in the feature addition process. Both user studies and perceptual metrics demonstrate that MaterialFusion outperforms existing approaches.

**Strengths:**

1. Effective combination of existing techniques. The proposed method integrates IP-Adapter for exemplar encoding and GaR for structural preservation. The design is simple yet effective, yielding clear improvements over baselines.

2. Clarity and motivation. The paper is well-written and well-motivated. Material transfer is a practical problem in AIGC. The authors articulate how existing methods struggle to balance material fidelity with structural consistency.

**Weaknesses:**

1. Limited novelty. The paper primarily adopts an “A+B” combination of existing components with minimal architectural innovation, offering limited new insights for the community. The masking idea is similar to previous work such as Blended Latent Diffusion.

2. Insufficient evaluation.

    1. Limited baselines. The comparisons are restricted to ZeST and GaR. This is insufficient given the breadth of recent developments. The paper should include comparisons with both specialized material transfer methods (e.g., MatSwap: Light-aware Material Transfers in Images) and general-purpose image editing approaches (e.g., Qwen-Image-Edit, FLUX.1 Kontext, Gemini).

    2. Lack of synthetic evaluation. No experiments on synthetic datasets are provided. Synthetic benchmarks with known ground truth would enable objective, pixel-level metrics such as PSNR and SSIM, strengthening the quantitative analysis.

3. The method is built upon Stable Diffusion 1.5, which is relatively outdated. It remains unclear whether the approach generalizes to newer architectures such as flow-matching models, potentially limiting its long-term applicability.

**Questions:**

1. How does the computational cost (inference time) compare with the baselines?

2. What is the resolution of images used in evaluation? How well does the method scale to high-resolution inputs?

---

### Official Review · Reviewer_9JvJ · 2025-10-29

**Soundness:** 3
**Presentation:** 3
**Contribution:** 2
**Rating:** 4
**Confidence:** 4

**Summary:**

This paper introduces MaterialFusion, a zero-shot approach for image material transfer editing. It combines IP-Adapter for image-based material prompting and GaR for preserving object geometry and details. The method also incorporates material transfer force and masking strategy to give finer grain material control. The results demonstrate promising zero-shot material transfer quality.

**Strengths:**

* The proposed training-free zero-shot editing by integrating IP-Adapter and GaR does show some promising material transfer results.
* The material transfer force is simple and effective for providing the progressive control over generated images.

**Weaknesses:**

Limited novelty, the overall pipeline seems to be an extended application of GaR with IP-Adapter for extra image prompting. The masking trick, in my opinion, is trivial in many diffusion editing methods. The transfer force is interesting but it is also pretty simple and straightforward. While I appreciate the simplicity of the method, the inherent novelty is also limited.

**Questions:**

* Why is dual-masking necessary? if only performing masking on the iteratively denoised latents (e.g., Appendix B) and without masking on cross-attention, does the method still perform well?
* Lack of comparison with some non training-free approaches (e.g., MARBLE, CVPR’25). It is good to include some comparisons.
* The base model SD1.5 in 2025 is a bit outdated. It is fine to compare against ZeST with the same SD1.5 base model. But can this method be seamlessly applied to recent DiT based models like FLUX? If so, how about comparing against some SoTA general image editing methods like (e.g., FLUX Kontext, Qwen-Image, Gemini image editing, etc.)?

---

### Official Review · Reviewer_z5Wr · 2025-11-02

**Soundness:** 3
**Presentation:** 3
**Contribution:** 2
**Rating:** 2
**Confidence:** 5

**Summary:**

This paper aims to perform exemplar-based material editing of an object in an input image and a material exemplar image.
The method builds upon the Guide-and-Rescale method, integrating IP-adapter to inject the material exemplar for transferring the material to the object which is identified using an object mask provided as input. The method employs a dual masking strategy, masking the object during cross-attention to only apply the material transfer to the object region and also at each denoising step to preserve the background during DDIM inversion. The method allows for controllable material transfer force using a scalar to control the extent of the material transfer. The method is evaluated against other methods on different real-world images. The evaluation uses quantitative metrics, LPIPS and CLIP. Furthermore, an extensive user study is provided to demonstrate improvements in material transfer and detail preservation.

**Strengths:**

- The method is well-motivated and the description is clear to be able to reproduce the results. The authors have provided the code which is well appreciated.
- The proposed method for masking is novel and motivated well to preserve the background during DDIM inversion and restricting the material transfer to the object region.
- The presentation of the results is clear and highlights the improvements over ZeST, GaR, and IP-adapter only approach.
- The user study is conducted with 100 participants to validate the claims of improved realism and detail preservation in the output images.

**Weaknesses:**

- One of the key limitations of the method is the dependency on the binary object mask. How robust is the method to the quality of the object mask? I believe that an analysis on this would be helpful.
- The results are only demonstrated on images with a single object. I suggest that the authors include some results on images of a scene with multiple objects. One interesting case would be a vase next to a cup with the material exemplar with high reflectivity, to see if the output image results in a reflection of the vase on the cup, and vice versa.
- L340: It would be helpful to illustrate the background changes as done in Figure 11 (supplementary section) in the main text to show why the second masking is necessary.
- The paper does not compare to MARBLE which allows for parametric material control.

**Questions:**

- Can the method perform material transfer on multiple objects of the same type in the input image? For example, if there are multiple instances of a cup with different materials in the input image with the same material exemplar, can the method transfer the material to all the cups consistently? This is a limitation based on Figure 9. If not, what causes this failure?
- How well does the method perform if the material exemplar is the same as the material of the object in the input image? Does it lead to producing the same image as the input image or something close to it?

---

### Official Review · Reviewer_QJje · 2025-11-02

**Soundness:** 3
**Presentation:** 3
**Contribution:** 2
**Rating:** 4
**Confidence:** 3

**Summary:**

The paper proposes MaterialFusion, a training-free material transfer framework that combines IP-Adapter, Guide-and-Rescale (GaR) , and a dual masking strategy to transfer material appearance from one image to an object in another. It also enables users to control the degree of the material transfer. The authors report superior perceptual quality and controllability compared to baselines like ZeST, IP-Adapter, and GaR alone.

**Strengths:**

* The choice of the architecture is justified and explained well
* Tackles an important and timely problem
* Strong results as shown in the paper
* The approach does not require additional training

**Weaknesses:**

* One of the main challenges addressed in this paper seems to be preserving the background. Their double masking approach was not well justified. Given that the target imagery seems to be a single object in the foreground, why not use a non-diffusion model approach such as cropping the object before material transfer and then adding back to the original background?
* The approach seems quite limited in terms of the types of textures that can be applied in this manner. It cannot handle more complex materials and or when the object's geometry is too complex
* There is no discussion or evaluation of accurately modeling lighting effects on the new material to mimic the lighting of the scene or the effect of new materials on the lighting of the surrounding. I assume this cannot be done with this approach, which is quite a significant limitation.
* Overall, the novelty is quite limited, essentially combining two existing techniques (IP-Adapter and GaR).

**Questions:**

Please comment on the weaknesses above

---

### Note · Authors · 2025-11-14

I have read and agree with the venue's withdrawal policy on behalf of myself and my co-authors.